# Ground-state oxygen holes and the metal–insulator transition in the negative charge-transfer rare-earth nickelates

Valentina Bisogni[1,2], Sara Catalano[3], Robert J. Green[4,5], Marta Gibert[3], Raoul Scherwitzl[3], Yaobo Huang[1,6], Vladimir N. Strocov[1], Pavlo Zubko[3,7], Shadi Balandeh[4], Jean-Marc Triscone[3], George Sawatzky[4,5] & Thorsten Schmitt[1]

The metal–insulator transition and the intriguing physical properties of rare-earth perovskite nickelates have attracted considerable attention in recent years. Nonetheless, a complete understanding of these materials remains elusive. Here we combine X-ray absorption and resonant inelastic X-ray scattering (RIXS) spectroscopies to resolve important aspects of the complex electronic structure of rare-earth nickelates, taking $NdNiO_3$ thin film as representative example. The unusual coexistence of bound and continuum excitations observed in the RIXS spectra provides strong evidence for abundant oxygen holes in the ground state of these materials. Using cluster calculations and Anderson impurity model interpretation, we show that distinct spectral signatures arise from a Ni $3d^8$ configuration along with holes in the oxygen $2p$ valence band, confirming suggestions that these materials do not obey a conventional positive charge-transfer picture, but instead exhibit a negative charge-transfer energy in line with recent models interpreting the metal–insulator transition in terms of bond disproportionation.

[1] Research Department Synchrotron Radiation and Nanotechnology, Paul Scherrer Institut, CH-5232 Villigen PSI, Switzerland. [2] National Synchrotron Light Source II, Brookhaven National Laboratory, Upton, New York 11973, USA. [3] Department of Quantum Matter Physics, University of Geneva, 24 Quai Ernest-Ansermet, 1211 Geneva 4, Switzerland. [4] Department of Physics and Astronomy, University of British Columbia, Vancouver, British Columbia, Canada V6T 1Z1. [5] Quantum Matter Institute, University of British Columbia, Vancouver, British Columbia, Canada V6T 1Z4. [6] Beijing National Laboratory for Condensed Matter Physics, and Institute of Physics, Chinese Academy of Sciences, Beijing 100190, China. [7] London Centre for Nanotechnology and Department of Physics and Astronomy, University College London, 17-19 Gordon Street, London WC1H 0HA, UK. Correspondence and requests for materials should be addressed to V.B. (bisogni@bnl.gov) or to T.S. (email: thorsten.schmitt@psi.ch).

The intriguing perovskite nickelates family ReNiO$_3$ (with Re = rare-earth)[1-4] have garnered significant research interest in recent years, due to the remarkable properties they exhibit. These include a sharp metal to insulator transition (MIT) tunable with the Re radius[2], unusual magnetic order[5] and the suggestion of charge order[6] in the insulating phase. While ReNiO$_3$ single crystals are very hard to synthesize, causing earlier experiments to be mainly restricted to powder samples, extremely high-quality epitaxial thin films can be now produced. As an additional asset, ReNiO$_3$ in thin film form can exhibit even richer properties compared to bulk, for example, tunability of the metal–insulator transition by strain[7,8], thickness[9-11] or even by ultrafast optical excitation of the substrate lattice degree of freedom[12]. In addition, there has been a lot of interest in nickelate-based heterostructures, motivated on one hand by theoretical predictions of possible superconductivity[13], and on the other hand by recent observations of exchange bias effects[14] and modulation of the orbital occupation due to strain and interface effects[15-18].

The origin of the rich physics behind these unique properties is complicated by the usual electron correlation problem of transition metal oxides[19]. As a consequence, a full understanding of the mechanism driving the MIT in ReNiO$_3$ remains elusive still today. Hampering the discovery of a universally accepted description of the MIT is the more fundamental problem of understanding the ReNiO$_3$ electronic structure and the corresponding Ni $3d$ orbital occupation.

In Fig. 1, we illustrate in a schematic representation of the single-electron excitation spectra how different electronic configurations can result from the two possible regimes of the effective charge-transfer energy $\Delta'$ (called for simplicity charge-transfer energy throughout the text). Using formal valence rules, it is expected that the Ni atoms exhibit a $3d^7$ (Ni$^{3+}$) character, likely in a low spin ($S = 1/2$) configuration. This ground state (GS) is obtained in Fig. 1a for $\Delta' > 0$. However, high-valence Ni$^{3+}$ systems are rare, and although many studies indeed view ReNiO$_3$ as conventional positive charge-transfer compounds yet with the addition of a strong Ni–O covalency and consequently a GS configuration of the type of $\alpha \cdot |3d^7\rangle + \beta \cdot |3d^8\underline{L}\rangle$ (where $\underline{L}$ is an O $2p$ hole)[1,20-23], mounting evidence suggests that the ground state disobeys conventional rules. Alternatively, a negative charge-transfer situation[24,25], where a finite density $n$ of holes $\underline{L}^n$ is self-doped into the O $2p$ band and Ni takes on a $3d^8$ configuration (that is, Ni $3d^8\underline{L}^n$), is recently receiving an increasing interest. This scenario is represented in Fig. 1b for $\Delta' < 0$. Notably, the negative charge-transfer picture is at the base of recent charge or bond disproportionation model theories where, as first suggested by Mizokawa[26], the disproportioned insulating state is characterized by alternating Ni $3d^8$ ($n = 0$) and Ni $3d^8\underline{L}^2$ ($n = 2$) sites arranged in a lattice with a breathing-mode distortion[27-30]. These models distinctively differ from the more traditional charge-disproportionation ones where the Ni $3d^7$ lattice moves towards an alternation of Ni $3d^6$ and $3d^8$ sites in the insulating phase[6,31-34].

To get a full understanding of the MIT and of the unique physical properties of the rare-earth nickelates, it is crucial to investigate the ground-state electronic structure in this class of materials. To this purpose, we stress that while both positive and negative charge-transfer interpretations introduced above can be described as highly covalent, there are striking inherent differences between the two. For the former[16,20,21,32,34], the GS $\alpha \cdot |3d^7\rangle + \beta \cdot |3d^8\underline{L}\rangle$ can be modelled as a Ni $3d^7$ impurity hybridizing with a full O $2p$ band. Here the primary low-energy charge fluctuations that couple to the GS in first order are the ones from the O $2p$ to the Ni $3d^7$ impurity, mixing in some Ni $3d^8\underline{L}$ and higher-order character into the wavefunction. However,

for the negative charge-transfer case[19,25-30], all Ni sites assume a $3d^8$ state with on average $n = 2$ holes in the six oxygens coordinating a central Ni ion. This case is more aptly modelled by a Ni $3d^8$ impurity hybridizing with a partially filled O $2p$ band. The electronic structure and consequently the character of the gap are vastly different in the two scenarios: $\alpha \cdot |3d^7\rangle + \beta \cdot |3d^8\underline{L}\rangle$ with a O $2p$–Ni $3d$-like gap or $3d^8\underline{L}^n$ with a O $2p$–O $2p$-like gap (refer to Fig. 1a,b, respectively).

Here we combine two X-ray spectroscopies, namely X-ray absorption (XAS) and resonant inelastic X-ray scattering (RIXS) at the Ni L$_3$-edge, to resolve if the electronic structure of ReNiO$_3$ follows a positive or negative charge-transfer picture. While the XAS results are similar to those previously reported, the first ever measurements of high-resolution Ni L$_3$ RIXS on NdNiO$_3$ provide crucial insights into the nature of the excitations present.

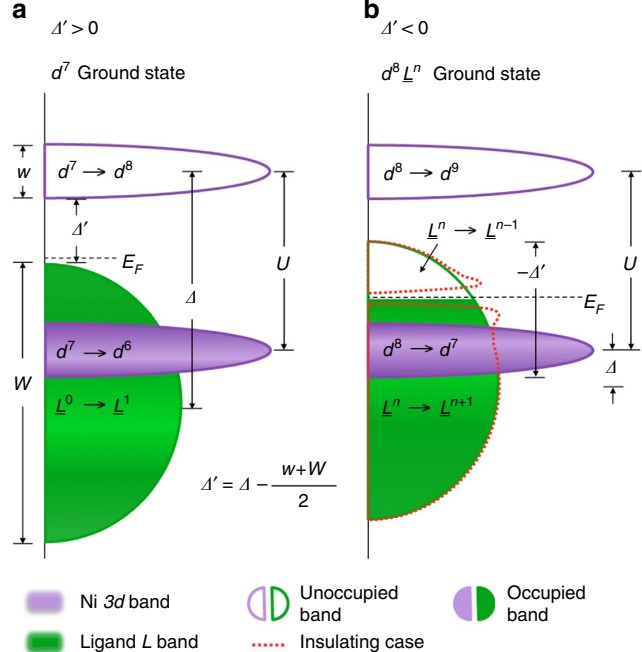

**Figure 1 | Single-electron excitation spectra in terms of charge removal and charge addition.** This sketch introduces conventional parameters used to describe charge dynamics in transition metal oxide (TMO) compounds with formal $3d^7$ filling: (1) charge-transfer energy $\Delta$: energy cost for transferring an electron/hole from the $\underline{L}$ band to the Ni $3d$ band (with respect to the band centre of mass); (2) Hubbard $U$: energy cost needed to remove an electron from the occupied $3d$ band and to add it to the unoccupied $3d$ band; (3) effective charge-transfer energy $\Delta'$: key parameter to distinguish between the two different regimes discussed here. $\Delta'$ is defined by the equation in the figure, starting from $\Delta$. This figure identifies the ground state (GS) and the gap character obtained for two $\Delta'$ regimes: (**a**) $\Delta' > 0$. In conventional positive charge-transfer compounds the lowest energy removal states are ligand based, and the lowest energy addition states are transition metal based, leading to a charge-transfer derived energy gap (O $2p$–Ni $3d$ like) and a $3d^7$ GS. (**b**) $\Delta' < 0$. In negative charge-transfer compounds, one hole per Ni is doped into the ligand band, giving a density of ligand holes $n$, $\underline{L}^n$. The GS is here Ni $3d^8\underline{L}^n$. The red-dashed contour bands are a cartoon-like demonstration of the opening of the gap in the mainly O $2p$ continuum resulting in the metal to insulator transition. In this case, the lowest energy removal and addition states are both ligand based, leading to an O $2p$–O $2p$ like gap. Under no circumstances can this type of gap result from configuration **a**, unless active doping is considered. Note that this figure neglects Ni–O hybridization, to provide clear distinction between the regimes.

The unusual coexistence of bound and continuum contributions across the narrow Ni $L_3$ resonance and the specific nature of the orbital excitations, allows us to verify that the electronic ground state contains abundant O $2p$ holes and that the Ni sites are indeed best described as Ni $3d^8 \underline{L}^n$, rather than a low spin Ni $3d^7$, showing that the ReNiO$_3$ are indeed self-doped, negative charge-transfer materials. Further, the RIXS spectra exhibit a clear suppression of the low-energy electron–hole pair continuum in the insulating phase, providing not only a fingerprint of the opening of the insulating gap at $T < T_{MI}$ but also experimental evidence of the dominant O $2p$-character for the states across $E_F$, as expected for a negative charge-transfer system.

## Results

**Bulk-like NdNiO$_3$ thin film.** Several high-quality NdNiO$_3$ thin films grown on a variety of substrates were investigated. Epitaxial films were prepared by off-axis radiofrequency magnetron sputtering[8,9,35,36] and were fully characterized by X-ray diffraction measurements, atomic force microscopy, transport and soft X-ray scattering measurements. In the following, we will focus on 30 nm thick NdNiO$_3$ film grown on (110)-oriented NdGaO$_3$ substrate under tensile strain conditions ($+1.6\%$ of strain) as a representative example of bulk ReNiO$_3$ in general. In this case, coupled metal–insulator and paramagnetic-to-antiferromagnetic transitions have been found at $T \sim 150$ K, consistent with the corresponding bulk compound[1].

**Bound and continuum excitations across Ni $L_3$ resonance.** XAS and RIXS measurements were carried out by exciting at the Ni $L_3$ edge, corresponding to the $2p_{3/2}$ to $3d$ electronic transition at around $\sim 852$ eV. The XAS spectra have been acquired in the partial fluorescence yield mode, by integrating the RIXS spectra for each incident photon energy $h\nu_{in}$ to insure the bulk sensitivity.

Figure 2a–c presents an overview of XAS and RIXS data for the 30 nm thick NdNiO$_3$ film, measured at both 300 K (metallic phase, red colour) and at 15 K (insulating phase, blue colour). The Ni $L_{2,3}$ XAS shown in Fig. 2a is in good agreement with the previously published data on NdNiO$_3$ (refs 20,21,37–39). At 15 K, the Ni $L_3$ region of the XAS (from 850 to 860 eV), is characterized by two clear structures—a sharp peak at 852.4 eV (A) and a broader peak at 854.3 eV (B)—both of which are present in other ReNiO$_3$ as well[1,16,20,24]. At 300 K both peaks are still recognizable, however, their separation is less evident.

A series of high-resolution RIXS spectra have been recorded across the Ni–$L_3$ resonance in steps of 0.1 eV, as shown in the intensity colour maps of Fig. 2b,c. Each spectrum obtained for a specific $h\nu_{in}$ measures the intensities of the emitted photons as a function of the energy loss $h\nu = h\nu_{in} - h\nu_{out}$, where $h\nu_{out}$ is the outgoing photon energy. RIXS is able to simultaneously probe excitations of diverse nature, for example, lattice, magnetic, orbital and charge excitations[40,41]. In addition, one can distinguish with RIXS between localized, bound excitations and delocalized excitations involving continua. For localized electronic excitations, the RIXS signal appears at a fixed $h\nu$ while scanning $h\nu_{in}$ across a corresponding resonance (Raman-like behaviour). Conversely, for delocalized electronic excitations involving continua, the RIXS signal has a constant $h\nu_{out}$ and therefore presents a linearly dispersing energy loss as a function of $h\nu_{in}$ (fluorescence-like behaviour)[40–43].

From the RIXS maps of the NdNiO$_3$ thin film in Fig. 2b,c one directly observes a clear, strong Raman-like response at around 1 eV of energy loss when tuning $h\nu_{in}$ to the XAS peak A. These atomic-like $dd$-orbital excitations, which are sensitive to the local ligand field symmetry, behave similarly to those observed in other oxide materials like the prototypical Ni $3d^8$ system NiO[44].

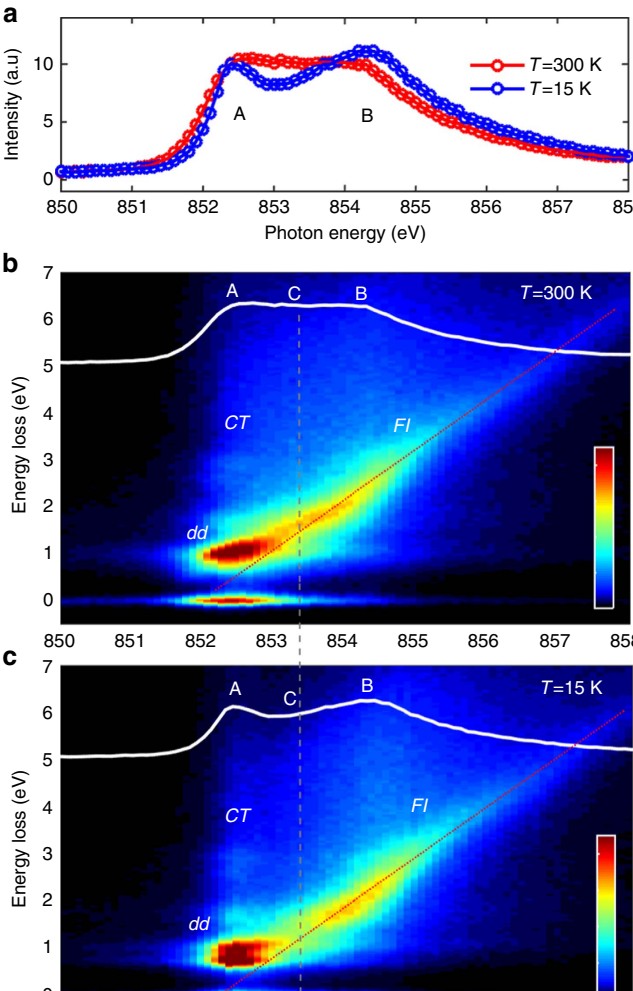

**Figure 2 | Overview of XAS and RIXS measurements for a 30 nm NdNiO$_3$ film on NdGdO$_3$.** (**a**) Ni $L_{3,2}$ XAS measured in partial fluorescence yield mode at 300 K (metallic phase) in red and at 15 K (insulating phase) in blue. Refer to Supplementary Note 2 and Supplementary Fig. 2 for XAS in total electron yield mode. (**b,c**) RIXS intensity map measured across the Ni $L_3$ edge at 300 K (15 K); intensity scale bar from 0 to 5 (a.u.). The white solid line displays the XAS measured at the same temperature. The letters A, B and C mark the three different incident energies mentioned in the text, while $dd$, $CT$ and $Fl$ refer to RIXS excitations of different character, also discussed in the text. The grey-dashed line indicates the incident energy giving the most pronounced changes in the RIXS map and in the XAS across the MIT. The red-dotted line provides a guide to the eye for the linearly energy dispersing $Fl$ feature.

A fluorescence-like contribution resonates instead at the XAS peak B, contrary to the Raman-like response dominated by multiplet effects observed in NiO at the corresponding Ni $L_3$ XAS shoulder[44]. Already by looking at the colour map, this fluorescence-like spectral signature is clearly visible all across the Ni $L_3$-edge and always with a linearly dispersing behaviour, as suggested by the red-dotted line overimposed to the data. Interestingly, the fluorescence intensity distribution in the NdNiO$_3$ RIXS map has also a strong temperature dependence, while at 300 K it merges continuously with the $dd$-excitations (Fig. 2b), at 15 K a dip in intensity is created corresponding to the incident photon energy C, $h\nu_{in} = 853$ eV (see Fig. 2c, dashed grey line).

To gain more insight into the origin of the observed Raman- and fluorescence-like spectral response, we closely examine the individual RIXS spectra and we perform a fitting analysis to extract the general behaviour of the main spectral components. As shown in Fig. 3a, the raw RIXS spectrum is decomposed into three different contributions (see Supplementary Note 1 and Supplementary Fig. 1). Referring to the photon energy $h\nu_{in} = A$, we identify from the corresponding RIXS spectrum localized $dd$-orbital excitations extending from 1 to 3 eV (black line), a broad background centred around 4 eV ($CT$, green line), and a residual spectral weight ($Fl$, magenta line) peaking at 0.7 eV between the elastic line and the dominating $dd$-profile.

Remarkably, at this photon energy even the fine multiplet structure of the $dd$-excitations is in good agreement with that of NiO[44], suggesting immediately that NdNiO$_3$ has an unusual Ni $3d^8$-like $S = 1$ local electronic structure similar to NiO. Figure 4

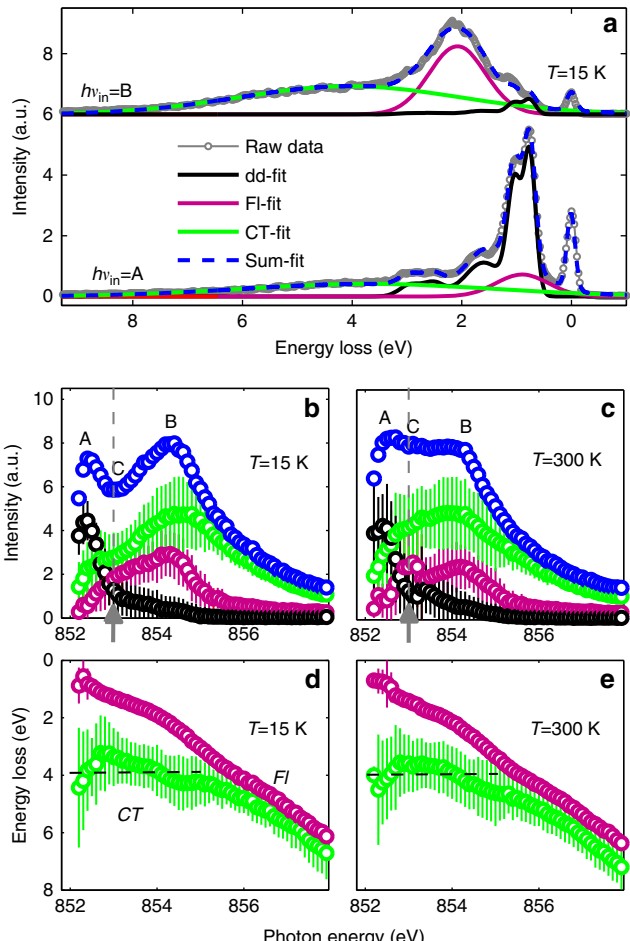

furthermore endorses this concept, displaying the comparison between the RIXS data and cluster model calculations performed for a Ni $3d^7$ GS (grey line, model parameters taken from ref. 39) and a crystal field calculation of the $dd$-excitations for a Ni $3d^8$ GS (black line, model parameters from NiO[45]; see also Supplementary Note 3 and Supplementary Table 1). The $dd$-excitations obtained for the Ni $3d^7$ case strongly differ from the present NdNiO$_3$ data not only in the peak energy positions displaced to higher energies, but also in the intensity distribution profile, thus ruling out a Ni $3d^7$ GS scenario. The $dd$-excitations calculated for the NiO in Ni $3d^8$ configuration instead present a remarkably good correspondence with the present data, verifying the $d^8$-like character of Ni in this compound: such a finding is the first key result of our study and directly poses the question if NdNiO$_3$ deviates from a conventional positive charge-transfer picture based on a Ni $3d^7$ GS in favour of a negative charge-transfer scenario based on a Ni $3d^8$ GS, as illustrated in Fig. 1b. Finally, we note that in the insulating phase an extra $dd$-peak emerges from the experimental data at ~0.75 eV, which is not captured by the Ni $3d^8$ calculation. We speculate that this contribution could be caused by symmetry-breaking phenomena (related to the presence of different Ni sites in the insulating phase). However, more advanced and detailed calculations have to be developed to reproduce this finding.

Referring now to $h\nu_{in} = B$ incident photon energy, the three contributions identified above for $h\nu_{in} = A$—$dd$, $CT$ and $Fl$—can be still distinguished in the corresponding RIXS spectrum. However, comparing the two spectra in Fig. 3a, we observe a shift in energy for the $CT$- and $Fl$-contributions, contrary to the $dd$-excitations which are fixed in energy loss, and a strong redistribution of spectral weight between the three spectral components.

**Figure 3 | RIXS data analysis across the MIT.** (**a**) RIXS line spectra (grey open-dot line) measured at $h\nu_{in} = A$ and $h\nu_{in} = B$ at 15 K. The thin solid lines refer to Gaussian fits of $dd$-excitations (black), the $CT$ excitation (green) and the $Fl$ excitation (magenta). The blue-dashed line refers to the sum of the three fitting contributions, plus the elastic line at 0 eV. (**b**,**c**) Integrated intensity of the fit $dd$-, $CT$- and $Fl$-excitations together with the total integrated intensity at 15 K (300 K). The grey arrow marks the incident energy giving the most pronounced changes in the RIXS integrated intensity across the MIT. (**d**,**e**) Peak energy dispersion for the $CT$- and $Fl$-excitation at 15 K (300 K). The same colour code as in **a** is used throughout the figure. The error bars of the model parameters are evaluated using the least square fitting routine and expressed in s.e.d. The error bars of the initial RIXS spectra were estimated assuming Poisson statistics.

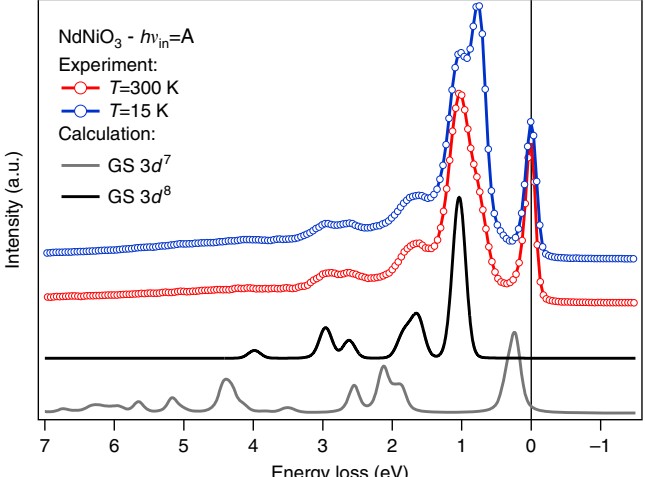

**Figure 4 | Comparison between experimental and calculated $dd$-excitations.** Data: RIXS spectra at $T = 15$ K (blue circles) and at $T = 300$ K (red circles), $h\nu_{in} = A$, $\sigma$ polarization. Calculation: crystal field RIXS calculation for a Ni $3d^8$ GS based on NiO parameters[45] (black line); cluster calculation for a Ni $3d^7$ GS using the parameters in ref. 39 (grey line). Please note that the elastic lines have been removed from the calculated spectra to better display the differences at the low energies. The experimental $dd$-line shape is well reproduced by the Ni $3d^8$ cluster calculation, supporting the negative charge-transfer scenario for ReNiO$_3$. The good matching between data and calculation shows that: (1) there is no inversion of crystal field in ReNiO$_3$ compared with NiO, in agreement with ref. 25 for dominating Coulomb interaction; (2) being the Ni–O distance shorter in ReNiO$_3$ than in NiO, the negative charge-transfer energy may lead to a reduction of the expected $t_{2g}$-$e_g$ splitting.

To disentangle localized versus delocalized character of the RIXS excitations, we extend in Fig. 3b–e the fitting analysis to all the RIXS spectra between 852 and 858 eV incident photon energies and we track for each contribution the integrated intensity (Fig. 3b,c) and the peak energy dispersion (Fig. 3d,e; see Supplementary Note 1 and Supplementary Fig. 1). While the $dd$-excitations (black dots) have a Raman-like behaviour throughout the energy range and clearly resonate at $hv_{in} = A$, the other two contributions ($CT$ in green and $Fl$ in magenta dots) resonate instead at $hv_{in} = B$ (see Fig. 3b, 15 K). Interestingly, and contrary to the similarity in their resonant behaviour, the $CT$- and the $Fl$-contributions present different peak energy dispersions as a function of the incident photon energy (Fig. 3d). The broad $CT$-peak (FWHM ~6 eV) displays a behaviour characteristic of well-known charge-transfer excitations between the central Ni site and the surrounding oxygens: constant energy loss (~4 eV) up to the resonant energy $hv_{in} = B$, and a fluorescence-like linear dispersion at higher incident photon energies. The $Fl$-peak (FWHM ~1 eV), instead, linearly disperses versus incident photon energy across the full Ni $L_3$ resonance: as introduced above, this behaviour is a clear fingerprint for a delocalized excitation involving continua. Overall, these findings are common to both low and high temperature data sets. However, in the metallic state (see Fig. 3c,e, 300 K) the intensity of both $CT$- and $Fl$-peaks is enhanced at $hv_{in} = C$: the resulting extra weight in the integrated RIXS intensity profile (Fig. 3c, blue dots) mimics the filling of the valley observed in the XAS spectrum at 300 K, and explains the absence of a dip in the intensity distribution of the RIXS map (Fig. 2b) at the same incident energy.

In addition, we examine the $Fl$-excitations more closely in Fig. 5, where we focus on the low energy loss range (<1.5 eV) of the high-statistics RIXS spectra obtained at the Ni $L_3$ pre-peak region, starting 1 eV before peak A. In Fig. 5a, we observe changes in the $dd$-excitations between 300 and 15 K (across the MIT) likely due in part to local rearrangements of the $NiO_6$ octahedra. Moreover, a sizeable spectral weight continuum around 0.2–0.5 eV (see Fig. 5a, inside the ellipse area) is present only at 300 K in the metallic phase, as better displayed by the RIXS colour maps for 300 and 15 K (see Fig. 5b,c, respectively).

**Anderson impurity model**. To understand the origin of the various RIXS excitations and their link to the electronic structure, one can employ an Anderson impurity model (AIM) interpretation (Supplementary Note 4). While the schematics in Fig. 1a,b show the single-particle removal and addition excitations for different charge-transfer scenarios, RIXS actually measures charge neutral excitations, which are well represented in a configuration interaction-based AIM. These charge neutral excitations are shown schematically in Fig. 6a for the negative charge-transfer case $\Delta' < 0$ and the positive charge-transfer case $\Delta' > 0$ (NiO like) of a Ni $3d^8$ impurity. As previously mentioned, for the $\Delta' > 0$ case, the only charge fluctuations possible in the AIM are from the full O $2p$ band to the Ni impurity level. While conserving the total charge, these fluctuations give rise to a Ni $3d^9\underline{L}$ band corresponding to charge-transfer excitations (shown in green in Fig. 6a for NiO). However, for the $\Delta' < 0$ case (recall Fig. 1b), the presence of a self-doped, partially filled O $2p$ density of states (DOS) extending across the Fermi level opens additional pathways for the neutral charge fluctuations. Electrons can either hop from the O $2p$ valence band to the O $2p$ conduction band leaving the Ni impurity occupation unchanged, yielding a characteristic low energy electron–hole pair continuum of $m$ excitations $d^8 v^m c^m$ marked in magenta ($\underline{v}$ is a hole in the valence band and $c$ an electron in the conduction band, as shown in the small O $2p$ DOS inset of Fig. 6a); or, to and from the Ni

impurity level, from the O $2p$ valence band and toward the O $2p$ conduction band, respectively, causing the impurity occupation to change by plus or minus one and yielding a charge-transfer like continuum of excitations at higher energy. Eventually these charge-transfer excitations can be dressed by associated electron–hole pair excitations, $v^m c^m$, resulting in $d^9 v^{m+1} c^m$ or $d^7 v^m c^{m+1}$ bands of excitations (green band). We stress that the here introduced low-energy electron–hole pair excitations can be obtained only for the $\Delta' < 0$ case, where the O $2p$ DOS crosses $E_F$.

The effects of these two impurity models on RIXS are detailed in Fig. 6b. The case of NiO can be solved numerically including the full correlations within the Ni $3d$ shell, and we show the calculated RIXS map in Fig. 6b. Comparing this to the schematic NiO configurations in Fig. 6a, we see that there are Raman-like $dd$-excitations below 4 eV, corresponding to reorganized Ni $3d^8$ orbital occupations, and a charge-transfer band at distinctively higher energy losses which is Raman-like for lower incident photon energies up to ca. 855 eV, before dispersing like fluorescence for higher photon energies. However, as the schematic in Fig. 6a suggests, the charge-transfer excitations do not extend down to low energy losses for the NiO case ($\Delta' > 0$). To gain further insight into the $NdNiO_3$ experimental data, the extracted $CT$- and $Fl$-dispersion curves of Fig. 3d are overlaid on the calculated RIXS map in Fig. 6b. Indeed, the $NdNiO_3$

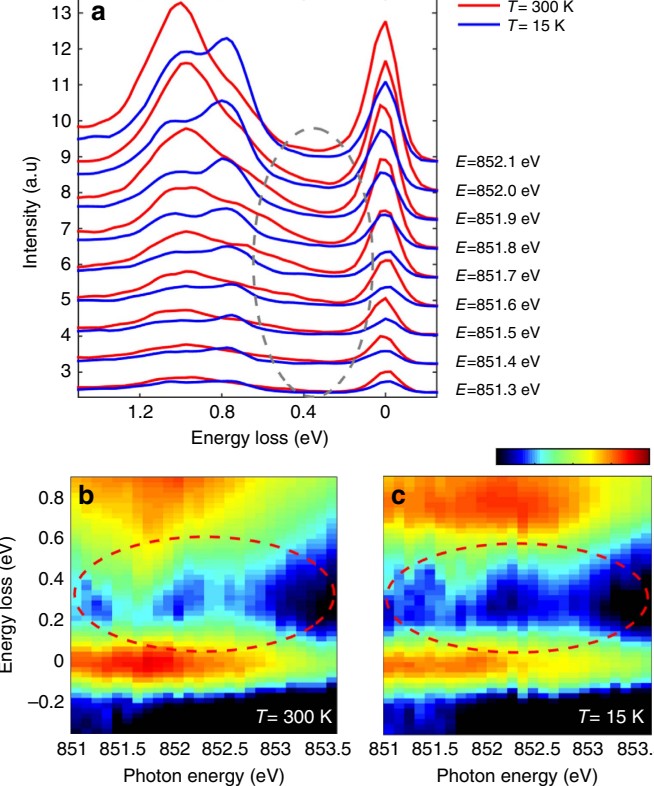

**Figure 5 | Low-energy electron–hole pair continuum in the RIXS spectra.**
(**a**) RIXS line spectra measured for incident energies going from $hv_{in} = A$-1 eV up to $hv_{in} = A$-0.2 eV in steps of 0.1 eV at 15 K (blue) and 300 K (red). Each spectrum has been acquired for 5 min. The grey ellipse highlights the energy loss region where the electron–hole pair continuum is more prominent.
(**b**,**c**), Magnification of the low energy loss region (<0.9 eV) of the RIXS map with a logarithmic intensity scale at 300 K (15 K); intensity scale bar from −3 to 1 (a.u.). The spectra have been normalized to the $dd$-area to have comparable background signal in the low energy loss region. The red ellipses underline the electron–hole pair continuum present at 300 K (**b**) and the intensity gap in the same energy window at 15 K (**c**).

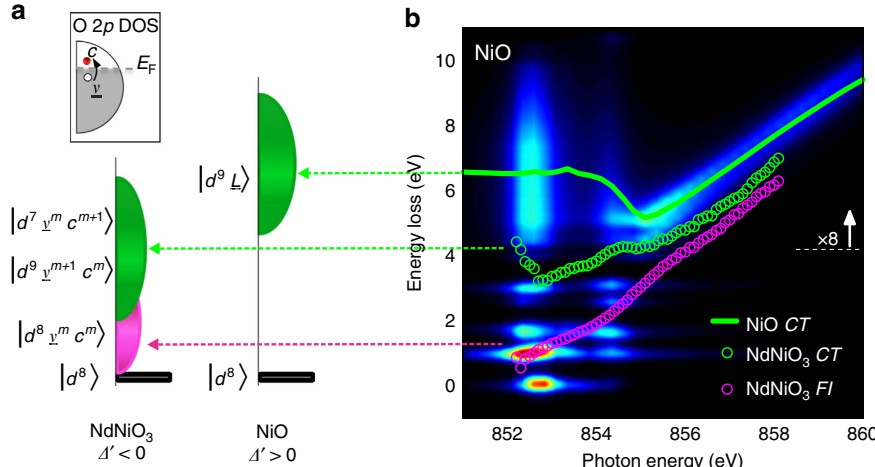

**Figure 6 | Anderson impurity model interpretation of the RIXS and XAS spectra. (a)** AIM schematic for charge neutral RIXS excitations. The configurations of the AIM are very different for $\Delta' < 0$ (NdNiO$_3$) and $\Delta' > 0$ (NiO) $3d^8$ compounds. CT-like excitations are obtained in both cases (green bands), while Fl-like excitations (magenta band) are found only for $\Delta' < 0$ and correspond to electron–hole pair excitations as shown in the O 2p DOS inset. **(b)** Calculated RIXS map for the positive charge-transfer compound NiO, using the AIM. The NdNiO$_3$ CT- and Fl-excitations dispersion curves from Fig. 3d are overlaid for comparison (green and magenta open dots, respectively), as well as the NiO CT-excitation dispersion (green solid line). Horizontal coloured lines make the connection between the RIXS excitations and the assigned interpretation in the AIM schematic of **a**.

CT-excitations (open green dots) show a similar behaviour as the NiO CT ones (solid green line). The NdNiO$_3$ Fl-excitations, instead, with their distinctive fluorescence-like dispersion differ from any of the NiO excitations. Interestingly, the identified Fl-contribution propagating down to very low energy losses is compatible instead with the electron–hole pair continuum excitations $d^8 \underline{v}^m c^m$ coming from the broad O 2p band: this finding is the second key result of our study and naturally occurs for the negative charge-transfer case ($\Delta' < 0$) as represented in the AIM schematic of Fig. 6a (magenta band).

## Discussion

The main findings of the presented data analysis and interpretation are as follows: localized dd-excitations sharply resonate at the XAS peak A with a lineshape consistent with a Ni $3d^8$ configuration; delocalized Fl-excitations mostly resonate at the XAS peak B and are interpreted as electron–hole pair excitations across the O 2p band cut by $E_F$ (Fig. 1b, green contours); spectral weight reduction of the electron–hole pair excitations close to zero energy loss (Fig. 5b,c) suggests the opening of an O–O insulating gap (Fig. 1b, red-dashed contours) at low temperature, in line with previously reported optical conductivity[46,47] studies and similarly as in more recent ARPES data[48] revealing a spectral weight transfer from near $E_F$ to higher binding energies across MIT.

This collection of results clearly identifies NdNiO$_3$ as a negative $\Delta'$ charge-transfer material, with a local Ni $3d^8$ configuration[49], a predominant O 2p character across the Fermi level[50] and a consequent GS of mainly Ni $3d^8\underline{L}^n$. This picture is compatible with the scenario proposed by Mizokawa[26], also discussed as bond disproportionation model in recent theoretical approaches[27–30,51,52], which comprises an expanded $3d^8$ Ni site ($n = 0$, $S = 1$) and a collapsed $3d^8\underline{L}^2$ Ni site ($n = 2$, $S = 0$) alternating in the insulating phase with the following spin order $\uparrow0\downarrow0$ and a homogeneous Ni $3d^8\underline{L}$ ($n = 1$) GS in the metallic phase. As underlined in previous works[28,29], this model is in agreement with several breakthrough experimental findings: (1/2 0 1/2) antiferromagnetic Bragg peak in the insulating phase[5]; charge ordering[6], which in this model is distributed among both Ni and O

sites instead of only Ni sites; absence of orbital order[34]; evidence of strong Ni–O covalence in the GS[4,20,21].

Furthermore, the different resonant behaviour extracted in this study for localized and delocalized RIXS excitations suggests that the two distinct XAS peaks marked at low temperature mostly result from the two different components of the GS, being XAS peak A mostly associated with a Ni $3d^8$ configuration, and XAS peak B with the delocalized ground-state Ni $3d^8\underline{L}^2$ configuration. This is in line with the energy dependence of the (1/2 0 1/2) peak resonating at $h\nu_{in} = $ A (refs 31,37), here assigned to the magnetically active $S = 1$ site.

In conclusion, by combining Ni L$_3$ XAS and RIXS measurements we studied the electronic ground-state properties of ReNiO$_3$, to discriminate the electronic structure between a negative and a positive charge-transfer scenario. By analysing the first ever high-resolution Ni L$_3$ RIXS data obtained for ReNiO$_3$, we identified the coexistence of bound, localized excitations and strong continuum excitations in both the XAS and the RIXS spectra, in contrast to earlier absorption studies which assumed primarily charge-transfer multiplet effects in the XAS. Further, we disentangled the continuum features in the RIXS spectra into charge-transfer and fluorescence excitations, showing the latter to arise due to the presence of a ground state containing holes in the oxygen 2p band. Electron–hole pair excitations from oxygen 2p states across the Fermi level have been identified down to zero energy loss, mimicking the opening of a gap for $T < T_{MI}$. All these experimental observations provide clear indication of an O 2p hole-rich ground state with Ni $3d^8\underline{L}^n$ electronic configuration as the main component, as expected for a negative charge-transfer system. This GS configuration lends support to the treatment of the ReNiO$_3$ as a $S = 1$ Kondo or Anderson lattice problem with a Ni $3d^8\underline{L}^n$ ($n = 1$) metallic GS, and realizing the MIT by a bond disproportionation leading to two Ni site environments: Ni $3d^8$ ($n = 0$, $S = 1$) and Ni $3d^8\underline{L}^2$ ($n = 2$, $S = 0$), differing in the hybridization with the O 2p hole states yet leaving the charge at the nickel sites almost equal. While this result is vital for the understanding of the rare-earth nickelate family per se, the combined XAS and RIXS approach demonstrated here opens the opportunity to classify the electronic structure for other cases of very small or negative charge-transfer gaps $\Delta'$, which could

be common to other materials with unconventionally high formal oxidation states (as for example sulfides, selenides and tellurides).

## Methods

**Experimental details.** XAS and high-resolution RIXS measurements have been performed at the ADRESS beamline of the Swiss Light Source[53], Paul Scherrer Institute. The sample has been oriented in grazing incidence, with the incoming photon beam impinging at 15° with respect to the sample surface. The scattering plane for the RIXS measurements was coinciding with the crystallographic $ac$-plane (or $bc$-plane, equivalently). All the data displayed here have been measured for incoming photons polarized parallel to the $a$ (b) axis (also referred to as $\sigma$ polarization, perpendicular to the scattering plane). For the RIXS measurements we used the SAXES spectrometer[54] prepared with a scattering angle of 130° and a total energy resolution of 110 meV. The spectrometer was set in the high-efficiency configuration, using the 1,500 lines per mm VLS spherical grating. This set-up allowed acquiring around 600–800 photons at the maxima of the prominent spectral structures already in 1 min. The recorded scattered photons were not filtered by the outgoing polarization.

**Data availability.** The XAS and RIXS data that support the findings of this study are available from the corresponding authors V.B. and T.S. upon request.

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

## Acknowledgements

This work was performed at the ADRESS beamline of the Swiss Light Source at the Paul Scherrer Institut, Switzerland. Part of this research has been funded by the Swiss National Science Foundation through its National Centre of Competence in Research MaNEP and the Sinergia network Mott Physics Beyond the Heisenberg (MPBH) model. The research leading to these results has received funding from the European Community's Seventh Framework Program (FP7/2007–2013) under Grant Agreement No. 290605 (COFUND: PSI-FELLOW) and ERC Grant Agreement No. 319286 (Q-MAC). Moreover, this work has received support through funding from the Canadian funding agencies NSERC, CRC, and the Max Planck/UBC center for Quantum Materials.

## Author contributions

V.B., S.C., M.G., R.S., G.S., J.-M.T. and T.S. planned the project. S.C, M.G., R.S., P.Z. and J.-M.T. grew and characterized the thin film samples. V.B., S.C., R.S., Y.H., V.S. and T.S. carried out the experiment. V.B. and T.S. analysed the data. R.G., S.B. and G.S. carried out the theoretical calculations. V.B., R.G., G.S. and T.S. wrote the paper with contributions from all authors.

## Additional information

**Competing financial interests:** The authors declare no competing financial interests.

