## [Peer Review File · Nature Communications]

Reviewers' comments:

Reviewer #1 (Remarks to the Author):

In this article the authors utilized high-resolution RIXS to explore the character of the electronic excitations in rare earth nickelates. Tracking the energy dependent nature of the resonant emission spectra, this work shows clear signatures of both localized and band-like excitations of the 3d states. Through theory, these results are used to explore the Ligand hole component in the electronic structure and to conclude that the groundstate has a large fraction of oxygen holes.

While this paper is very important and contains very solid science, there are some points that need to be clarified before it is ready for publication in Nature Communications.

1.) The concept of Nickelates as a negative charge transfer material has been discussed before in many papers, which was duly noted by the authors. I would say that the unique finding here clearly seeing the localized 3d⁸ component of the electronic structure, which is really only possible if viewing this system as 3d⁸L as opposed to 3d⁷. However, the authors have several discussions about how this work proves $\Delta < 0$ except I think previous work already put the community in that perspective.

2.) It is surprising that there are no oxygen K edge results to compliment this finding. The results of XAS to date have often pointed to the large 3d-2p prepeak as a signature of the state having a large fraction of Ligand hole states. While they can track the Ligand hole part through the CT feature in the 3d spectra, it should be important to look at this from the perspective of the oxygen 2p states. If the system is a really self doped, wouldn't there be a strong signature in the 2p states as well?

3.) In Fig. 4, why is there a single peak in the calculation but a clear doublet in the insulating phase? Is this a strain effect maybe? What is missing? Also, this doublet collapses in the metallic phase. What does this tell us about how the electronic structure is changing? There is clearly still a localized component even in the metallic phase.

4.) The fitting of the FI component is not completely convincing. Since it rides underneath several other peaks, how do the authors trust the peak shape and intensity? There could be a large error associated with the feature details. This is a key feature since the work associates it with the delocalized component.

Reviewer #2 (Remarks to the Author):

Bisogni et al. report on "Ground state oxygen holes and the metal-insulator transition in the negative charge transfer rare-earth nickelates." The authors performed soft X-Ray RIXS measurements and AIM calculations to unravel the electronic nature of NdNiO₃ through detecting the energy dispersion of CT excitations and the continuum of electron-hole excitations. The major results are 1) The comparison of dd excitations with calculations shown in the paper gives evidence for Ni d⁸ character. 2) With the decomposition of RIXS profile to dd, CT and FI components, the FI energy dispersion suggests that the ground state configuration exhibits a negative charge transfer energy, disobeying conventional rules. Overall this manuscript is well written and is an important study. I suggest its publication if the following suggestions are taken into consideration.

(1) One weak point of the RIXS data analysis shown in the paper is that the uncertainty of parameters for the decomposition into dd, CT, and FI profiles are large. The conclusion of negative charge transfer

is presently drawn from these data fittings. In fact, this conclusion does not critically depend on the values of fitting parameters. The authors are suggested to qualitatively discuss the energy dispersion of the broad feature of the continuum of electron-hole excitations before resorting to data fitting.

(2). In Fig 2(c), the dashed grey line does line with 853 eV, where the "gap" occurs. Figure 2(c) is inconsistent with the text.

(3) There are three major findings drawn in the discussion on p11. The referee fails to understand that Fig. 5 supports the third one "iii) spectral weight suppression of the electron-hole pair excitations close to zero," as these low-energy RIXS might result from several contributions such such as multimagnons, orbital excitations, and maybe multiphonons. More analyses are needed to draw the above conclusion.

(4) Some information about RIXS measurements are missing, e.g. the polarization of incident and scattered X-rays, and sample orientation etc.

Reviewer #3 (Remarks to the Author):

The electronic structure of the rare-earth nickelates remains unresolved, hampering the a full understanding of the mechanisms behind the metal-insulator transition (MIT). Understanding the electronic ground state of these materials is crucial for advancing our understanding of electronic correlations in solids. Using x-ray absorption and resonant inelastic X-ray scattering the authors study the electronic structure of NdNiO₃. They find an unusual coexistence of bound and continuum excitations, which they attribute to holes in the O 2p states, i.e., Ni 3d⁸ configuration with holes in the oxygen 2p valence band. From this picture it is concluded that the rare-earth nickelates do not obey a "conventional" positive charge-transfer picture, but instead exhibit a negative charge-transfer energy. The results are interesting and warrant publication in Nature Communications. However, the authors should clarify the following issue: Figure 1 represents two distinct ground state configurations: on left, the Ni d⁷ and, on the right, Ni d⁸ with holes in the oxygen 2p band. In most oxides, the O 2p bands are quite low in energy, with a typical ionization potential of more than 7 eV (see Xu and Schoonen, American Mineralogist, Volume 85, pages 543-556, 2000). That would imply that the ionization energy for the Ni state in the rare-earth nickelates is more than 7 eV since holes would be in the O 2p states. Could the authors comment on this quite high ionization energy? How does this picture fit in when looking at the ionization energies of the neighboring 3d metals, such as Co and Cu. Can the authors reconcile their results using cluster and Anderson impurity model calculations with those of first-principles methods such as DFT/DFT+U or DMFT (see Park et al., Phys. Rev. B 90, 235103 (2014)).

REVIEWERS' COMMENTS:

Reviewer #1 (Remarks to the Author):

In this article the authors utilized high-resolution RIXS to explore the character of the electronic excitations in rare earth nickelates. Tracking the energy dependent nature of the resonant emission spectra, this work shows clear signatures of both localized and band-like excitations of the 3d states. Through theory, these results are used to explore the Ligand hole component in the electronic structure and to conclude that the groundstate has a large fraction of oxygen holes.

This work represents a nice piece of science and the authors have addressed my concerns. The paper is now ready for publication in Nature Communications.

Reviewer #3 (Remarks to the Author):

I am quite satisfied with the authors response to the referees. The experiments reveal important features of the electronic structure and physics of nickelates, bringing a substantial advance to the field of correlated oxides.

Answers to Reviewers' comments on NCOMMS-16-05964

Reviewer #1 (Remarks to the Author):

In this article the authors utilized high-resolution RIXS to explore the character of the electronic excitations in rare earth nickelates. Tracking the energy dependent nature of the resonant emission spectra, this work shows clear signatures of both localized and band-like excitations of the 3d states. Through theory, these results are used to explore the Ligand hole component in the electronic structure and to conclude that the groundstate has a large fraction of oxygen holes.

While this paper is very important and contains very solid science, there are some points that need to be clarified before it is ready for publication in Nature Communications.

[Our reply]: We thank the reviewer for his/her careful reading of our manuscript and for acknowledging its "importance" and the "very solid science". We also appreciate his/her constructive comments, which we address below in details.

1.) The concept of Nickelates as a negative charge transfer material has been discussed before in many papers, which was duly noted by the authors. I would say that the unique finding here clearly seeing the localized 3d⁸ component of the electronic structure, which is really only possible if viewing this system as 3d⁸L as opposed to 3d⁷. However, the authors have several discussions about how this work proves $\Delta < 0$ except I think previous work already put the community in that perspective.

[Our reply]: We thank the reviewer for recognizing the new input of our study in terms of the local 3d⁸ electronic structure of the Ni site. This result in combination with our second experimental finding of abundant holes in the oxygen states (also revealed by other studies, as cited in the manuscript) are both essential to reach a definitive understanding of the ground state configuration in ReNiO₃ based on experimental evidence: we identify in this way the 3d⁸L electronic configuration as the main component in the ground state configuration of rare earth nickelates. Furthermore, such a ground state can be realized only at the presence of negative charge transfer energy defined relative to the top of the oxygen band. Therefore, our work provides an unambiguous experimental evidence for this scenario. At the same time, we agree with the reviewer that the negative charge transfer scenario was already discussed within the rare earth nickelates community mostly from the theoretical point of view. In this respect, our study provides a solid experimental answer to the ongoing debate about how one should start in describing the electronic ground state in this family of materials, which sees still today many papers published in high profile journals referring to a 3d⁷ electronic configuration with Ni³⁺ as the main component of the ground state (or in other words with a positive charge transfer energy as defined above) instead of a 3d⁸L state with Ni²⁺. The reason behind this is that in the description of the physics of the rare earth nickelates it is of extreme importance to start with the correct configuration before switching on the Ni 3d - O 2p hybridization. In fact, we underline that if we start with a 3d⁷ electronic configuration we are actually starting with a gapped system, while if we start with a 3d⁸L configuration, we are starting instead with a metallic state, having holes in the oxygen 2p band. As a proof of this on-going discussion, we list below the most representative works published in the last 10 years on these compounds, which indeed alternate interpretations based on a ground state of mostly 3d⁷ character [R1-2, R4-7,

R12-13] to the $3d^8$ one [R3, R8-11, R14-17].

References

- R1. V. Scagnoli et al., PRB 73, 100409 (2006)
- R2. K. Horiba et al., PRB 76, 155104 (2007)
- R3. I. Mazin et al., PRL 98, 176406 (2007)
- R4. M. Medarde et al., PRB 80, 245105 (2009)
- R5. E. Benckiser et al., Nat. Materials 10, 189-193 (2011)
- R6. Y. Bodenthin et al., J. Phys.: Condens. Matter 23, 036002 (2011)
- R7. J. Chaloupka et al., PRL 100, 016404 (2011)
- R8. S. Lee et al., PRL 106, 016405 (2011)
- R9. H. Park et al., PRL 109, 156402 (2012)
- R10. B. Lau et al., PRL 110, 126404 (2013)
- R11. J. Liu et al., Nat. Comm. 3714 (2013)
- R12. M. Wu et al., PRB B 88, 125124 (2013)
- R13. J. A. Alonso, et al., PRB 87, 184111 (2013)
- R14. S. Johnston et al., PRL 122, 106404 (2014)
- R15. R. Jaramillo et al., Nat. Physics 10, 304 (2014)
- R16. A. Subedi et al., PRB 91, 075128 (2015)
- R17. J. W. Freeland et al., J. Electron. Spectrosc. Relat. Phenom. 208, 56 (2016)

2.) It is surprising that there are no oxygen K edge results to compliment this finding. The results of XAS to date have often pointed to the large 3d-2p prepeak as a signature of the state having a large fraction of Ligand hole states. While they can track the Ligand hole part through the CT feature in the 3d spectra, it should be important to look at this from the perspective of the oxygen 2p states. If the system is a really self doped, wouldn't there be a strong signature in the 2p states as well?

[Our reply]: The reviewer correctly refers to the presence of a very strong prepeak in the O K XAS spectra of rare earth nickelates, which is usually interpreted as proportional to the oxygen holes in the material given by the very strong Ni 3d – O 2p hybridization. We also did XAS measurements at O K edge, both in the metallic and in the insulating phase. We obtained spectra in good agreement with measurements already published in the literature either from bulk samples [PRB 46, 14975 (1992), J. Electron. Spectrosc. Relat. Phenom. 208, 56 (2016)] or thin films [Nat. Comm. 3714 (2013)]. In order to respond to the reviewer's comment, we added now our O K XAS spectra in the Supplementary Information, to show consistency between our measurements and the literature. However, we do not discuss these measurements in the main text. In fact, while the usual interpretation of the O K XAS prepeak is essentially based on the strong hybridization of O 2p with Ni 3d states, it may also be that a close study of this prepeak could also provide insight into whether the charge transfer energy is positive or negative. However, the conclusions from such an analysis would likely not be as clear as from our Ni L_3 plus impurity model analysis, because the O edge is not so clearly divided into bound and continuum excitations. Further, the edge lacks the presence of dd -excitations, which can clearly distinguish the valence of the elements of interest (whereas our Ni L_3 edge data shows a clear presence of d^8 rather than d^7 multiplets). This is the reason we focus our present study on the Ni L_3 edge, but the reviewer is correct in that future studies could also approach the problem from the O K edge resonance (although they would likely be more complicated studies and

might not lead to as strong of conclusions).

3.) In Fig. 4, why is there a single peak in the calculation but a clear doublet in the insulating phase? Is this a strain effect maybe? What is missing? Also, this doublet collapses in the metallic phase. What does this tell us about how the electronic structure is changing? There is clearly still a localized component even in the metallic phase.

[Our reply]: The reviewer raises a very interesting question. In the caption of Fig. 4 we were suggesting that the presence of two different Ni sites could be a possible explanation for the extra peak observed in the *dd*-excitations around 0.75 eV at low temperature (insulating phase). However, at this stage, we do not really understand everything about this rich and complicated set of data yet. In fact, in the metallic phase, we observed instead a polarization dependence in the XAS around the Ni L₃ pre-edge and correspondingly, exclusively for this excitation energy, the RIXS spectra also display the extra *dd*-peak \sim 0.75 eV, but with a strong polarization dependence. This *dd*-peak then gets quickly suppressed (for both polarizations) at higher excitation energies as displayed in Fig. 4 for the metallic phase, contrary to what observed in the insulating phase where the 0.75eV *dd*-peak is polarization- and energy-independent. Up to now we do have evidence of some ongoing symmetry breaking, however it is pretty difficult at this stage to discuss its origin. A more detailed theoretical approach would be required to tackle this problem that should go even far beyond a single-site DMFT or a two-site cluster model. This approach would be helpful to sort out what the consequences are when considering a negative charge transfer system like the rare earth nickelates, instead of a standard positive charge transfer system or a mixed valence one.

4.) The fitting of the FI component is not completely is convincing. Since it rides underneath several other peaks, how do the authors trust the peak shape and intensity? There could be a large error associated with the feature details. This a key feature since the work associates it with the delocalized component.

[Our reply]: The reviewer correctly says that the '*FI*' component rides underneath other peaks in the RIXS spectrum. This happens in particular for excitation energies around peak A in the XAS, between 852 eV to 853 eV. In this energy range the *dd*-excitations are the dominant component of the spectrum and have been fit at $h\nu=A$ with four Gaussian-curves, which capture their shape very accurately. The overall *dd*-profile is then kept fixed throughout the fitting of the RIXS spectra as a function of excitation energy (852 eV – 858 eV), while its amplitude is a free fitting parameter. Two other Gaussian curves with three free parameters each are used to fit the data (not considering the elastic peak). The error bars of the fit analysis are estimated with the least squares method and displayed in Fig. 4 (b-e) with the corresponding fit parameters (intensity and peak position) of the two Gaussians named '*CT*' and '*FI*'. We underline that, despite the absence of constraints in the fitting procedure, not only the extracted peak position values follow a very smooth and correlated trend as a function of the excitation energy – including the energy region around 852-853 eV – but also: 1) the intensity values trace a clear resonance at Peak B of the XAS in agreement with the delocalized nature of the *CT* and *FI* features; 2) the FWHM values remain constant within $\pm 10\%$ in the full energy range [see Fig. S1 (c)]; 3) the error bars are considerably small, thus supporting the good quality and reliability of the displayed

fitting analysis. In our previous version of the manuscript, we presented the details of the fitting analysis in the Supplementary Information. In view of the concern of the reviewer and the relevance of this analysis for the messages discussed in the main text, we have now improved the explanation of our fitting procedure. We also stress that this fitting analysis should not be interpreted as an exact decomposition of the individual spectral contributions, but as a general description used to capture the behaviour of the main features observed in the RIXS data of this complicated system.

Reviewer #2 (Remarks to the Author):

Bisogni et al. report on "Ground state oxygen holes and the metal-insulator transition in the negative charge transfer rare-earth nickelates." The authors performed soft X-Ray RIXS measurements and AIM calculations to unravel the electronic nature of NdNiO₃ through detecting the energy dispersion of CT excitations and the continuum of electron-hole excitations. The major results are 1) The comparison of dd excitations with calculations shown in the paper gives evidence for Ni d⁸ character. 2) With the decomposition of RIXS profile to dd, CT and FI components, the FI energy dispersion suggests that the ground state configuration exhibits a negative charge transfer energy, disobeying conventional rules. Overall this manuscript is well written and is an important study. I suggest its publication if the following suggestions are taken into consideration.

[Our reply]: We thank the reviewer for his/her careful reading of our manuscript and for acknowledging its quality and importance. We also appreciate his/her constructive comments, which we address below in details.

(1) One weak point of the RIXS data analysis shown in the paper is that the uncertainty of parameters for the decomposition into dd, CT, and FI profiles are large. The conclusion of negative charge transfer is presently drawn from these data fittings. In fact, this conclusion does not critically depend on the values of fitting parameters. The authors are suggested to qualitatively discuss the energy dispersion of the broad feature of the continuum of electron-hole excitations before resorting to data fitting.

[Our reply]: We thank the reviewer for his/her comment. We modified the main text accordingly by underlining the unusual appearance of an energy dispersing broad feature in the raw data still responding to excitation energies around the pre-edge region of the Ni L₃ XAS. This finding in fact suggests *per-se* the presence of a broad band in the vicinity of the Fermi level even in the insulating phase. We clarified in the main text that the presented fitting analysis doesn't provide quantitative information, but it rather represents a systematic approach for capturing the general behavior of the main spectral features observed in the RIXS spectra of the rare earth nickelates.

(2). In Fig 2(c), the dashed grey line does line with 853 eV, where the "gap" occurs. Figure 2(c) is inconsistent with the text.

[Our reply]: The reviewer's criticism has been very helpful here for identifying an inappropriate notation used on our side in relation to Figure 2(c), which was misleading for readers and the reviewer. The purpose of the dashed grey line was to highlight the presence of a dip in intensity happening in the experimental RIXS map at around 853 eV, when the system is in the insulating phase. This dip in intensity was naively called 'gap' in our previous version just because of its appearance in the 2D color map. However, this doesn't represent a direct fingerprint of the opening of the insulating gap. We modified therefore the current version of the paper by replacing 'gap' with 'dip in intensity'.

(3) There are three major findings drawn in the discussion on p11. The referee fails to understand that Fig. 5 supports the third one "iii) spectral weight suppression of the electron-hole pair excitations close to zero.," as these low-energy RIXS might result from several contributions such as multimagnons, orbital excitations, and maybe multiphonons. More analyses are needed to draw the above conclusion.

[Our reply]: The reviewer is correct in saying that more analyses are needed to draw a firm conclusion. However, this has to wait new experimental opportunities with better experimental resolution and/or higher efficiency to discriminate the presence of the residual spectral weight at low energy losses. Unfortunately, the present data do not allow the rigor needed to make a definitive assignment. Thanks to the reviewer's criticism, we realized that our statement was not correctly formulated and we modified it by relieving the assertive tone and rephrasing it as a 'possible interpretation'. Such an interpretation is in any case supported by several arguments: 1) the absence of dd -excitations in the low energy loss range (<0.5 eV), according to the $3d^8$ local electronic structure for the Ni; 2) the expected energy scale for magnetic excitation being well below the current data energy resolution (100 meV); 3) although the strength of multimagnon modes or multiphonon modes in the RIXS spectra cannot be quantified in details, however, these modes are expected to be enhanced in the insulating/magnetic phase where the low-lying electron-hole excitations are damped because of the gap opening. Given the strong intensity reduction observed in the RIXS map at low energy losses while going to the insulating/magnetic phase, our understanding is that any multimagnon and multiphonon feature would simply be too weak to be seen. Therefore we interpret the spectral weight reduction observed across $T_{MI/AFM}$ in Fig. 5 (b-c) as related to electron-hole excitations reduction in intensity caused by the opening of the gap.

(4) Some information about RIXS measurements are missing, e.g. the polarization of incident and scattered X-rays, and sample orientation etc.

[Our reply]: We thank the reviewer for pointing out that the information related to the experimental geometry were not easily accessible. Therefore, we gathered them from the caption of Fig. 2 and Sec. 'Bound and Continuum excitations across Ni L_3 resonance' into a unique paragraph under the 'Methods -- Experimental details' Section.

Reviewer #3 (Remarks to the Author):

The electronic structure of the rare-earth nickelates remains unresolved, hampering the a full

understanding of the mechanisms behind the metal-insulator transition (MIT). Understanding the electronic ground state of these materials is crucial for advancing our understanding of electronic correlations in solids. Using x-ray absorption and resonant inelastic X-ray scattering the authors study the electronic structure of NdNiO₃. They find an unusual coexistence of bound and continuum excitations, which they attribute to holes in the O 2p states, i.e., Ni 3d⁸ configuration with holes in the oxygen 2p valence band. From this picture it is concluded that the rare-earth nickelates do not obey a "conventional" positive charge-transfer picture, but instead exhibit a negative charge-transfer energy. The results are interesting and warrant publication in Nature Communications. However, the authors should clarify the following issue: Figure 1 represents two distinct ground state configurations: on left, the Ni d⁷ and, on the right, Ni d⁸ with holes in the oxygen 2p band. In most oxides, the O 2p bands are quite low in energy, with a typical ionization potential of more than 7 eV (see Xu and Schoonen, American Mineralogist, Volume 85, pages 543-556, 2000). That would imply that the ionization energy for the Ni state in the rare-earth nickelates is more than 7 eV since holes would be in the O 2p states. Could the authors comment on this quite high ionization energy? How does this picture fit in when looking at the ionization energies of the neighboring 3d metals, such as Co and Cu. Can the authors reconcile their results using cluster and Anderson impurity model calculations with those of first-principles methods such as DFT/DFT+U or DMFT (see Park et al., Phys. Rev. B 90, 235103 (2014)).

[Our reply]: We thank the referee for noting that our results are interesting and that they warrant publication in Nature Communications. Concerning the questions regarding the energetics of Figure 1 in the manuscript, we are happy to provide clarification.

Generally, oxides near the later end of the transition metal series have larger values of Coulomb repulsion (U) and larger electron affinities of the transition metal ions [R1, R2]. In addition, increasing the oxidation state near the end of the series also strongly increases the electron affinity of the transition metal ion, keeping the ionization potential of the O 2p approximately the same. Note that in these terms, the charge transfer energy is basically the ionization potential of the oxygen minus the electron affinity of the transition metal, where in the case of the rare earth nickelates the transition metal would be Ni³⁺. Having a higher valence, Ni³⁺ provides a much more attractive potential for electron addition than Ni²⁺ so the electron affinity is much larger (in fact, in free space it is about U larger). Given that NiO is already established as a positive charge transfer system, an increase in oxidation state (going from Ni²⁺ in NiO to Ni³⁺ for the rare earth nickelates) would move the electron addition state (upper Hubbard band of Fig. 1a in the manuscript) down by such a large amount that it would be below the top of the oxygen band, resulting in a reshuffling of charge and making Ni²⁺ with holes in the oxygen band (i.e. negative charge transfer) a better starting point. Of course in real solids things are more complicated due to the influence of factors such as the Madelung potential, therefore it is crucial to have experimental insight into whether a particular material is in the positive or negative charge transfer regime.

In considering the neighboring 3d-metals Co and Cu as suggested by the reviewer, we expect the aforementioned trend for the 3d-series to hold true. For comparable Co³⁺ oxides like LaCoO₃, the slightly smaller electron affinity and Coulomb repulsion of Co compared to Ni suggest that it is likely a positive charge transfer system, and experiments support this suggestion [R3]. For Cu³⁺ oxides, however, we expect an even larger electron affinity and thus a negative charge transfer behavior. Indeed, this has been found experimentally for NaCuO₂ [R4]. Thus Ni³⁺ is in between these two cases, and our work provides the crucial experimental demonstration of

negative charge transfer energetics for the rare earth nickelates. Note that for tetravalent oxides electron affinities will be even higher, such that tetravalent oxides from Cu down to Fe might all also be negative charge transfer systems.

Finally, regarding the last comment of the referee, we note that our experimental results and corresponding theoretical interpretations are consistent with the DFT+DMFT study cited by the referee [R5], and with the related studies cited in our manuscript [R6-R9]. These theoretical studies assume a priori a $3d^8$ Ni occupation, and from that assumption they find the unique bond disproportionation phenomenon to describe the metal-insulator transition. Our work provides the crucial link to experiment, showing that such a $3d^8$ Ni occupancy is indeed a much better starting point for theoretical work, because of the required presence of a finite density of oxygen 2p holes (corresponding to on average two holes per octahedron).

References

- R1. Phys. Rev. B 46, 3771 (1992)
 - R2. Phys. Rev. Lett. 55, 418 (1985)
 - R3. Phys. Rev. Lett. 97, 176405 (2006)
 - R4. Phys. Rev. Lett. 67, 1638 (1991)
 - R5. Phys. Rev. B 90, 235103 (2014)
 - R6. Phys. Rev. Lett. 109, 156402 (2012)
 - R7. Phys. Rev. Lett. 110, 126404 (2013)
 - R8. Phys. Rev. Lett. 112, 106404 (2014)
 - R9. Phys. Rev. B 91, 075128 (2015)
-

List of changes:

- 1) We ensured compliance to Nature Communications styles (optimized title of the figures, added a Methods section).
- 2) In response to Reviewer 1 – point 2, we added O K edge XAS data and description in the Sec. A of the Supplementary Information.
- 3) In response to Reviewer 1 – point 3, we added a sentence about the speculative origin of the 0.75 eV peak in the main text and removed the corresponding explanations from the Caption of Fig. 4.
- 4) In response to Reviewer 1 – point 4, we improved the explanation of the RIXS Spectra fitting analysis in the Supplementary Information.
- 5) In response to Reviewer 2 – point 1, we modified the text at Page 7 to expand the discussion of the Fluorescence feature observed in Fig. 2 (b-c). To this purpose we modified Fig. 2 with the addition of a red-dotted line and we added a sentence in the main text “Already by looking at the colour map, this feature seems to be present all across the Ni L_3 edge and always with a linearly dispersing behaviour, as suggested by the red dotted line over imposed to the data”. Also, we added an extra sentence in the following paragraph to clarify the role of the fitting

analysis “and we perform a fitting analysis to extract the general behaviour of the main spectral components”.

6) In response to Reviewer 2 – point 2, we modified a sentence in the text at Page 7 “at 15 K a gap in intensity opens corresponding to the incident photon energy C , $h\nu_{\text{in}} = 853 \text{ eV}$ ” with “at 15 K a dip in intensity is created corresponding to the incident photon energy C , $h\nu_{\text{in}} = 853 \text{ eV}$ ”.

7) In response to Reviewer 2 – point 3, we modified a sentence in the text at Page 11 “spectral weight suppression of the electron-hole ... indicates the opening of an O-O insulating gap” with “spectral weight reduction of the electron-hole ... suggests the opening of an O-O insulating gap”.

8) In response to Reviewer 2 – point 4, we added in the Methods section a paragraph about the “Experimental Details” and removed the information previously distributed between the main text and Fig. 2 caption.

Answers to Reviewers' comments on NCOMMS-16-05964A

Reviewer #1 (Remarks to the Author):

In this article the authors utilized high-resolution RIXS to explore the character of the electronic excitations in rare earth nickelates. Tracking the energy dependent nature of the resonant emission spectra, this work shows clear signatures of both localized and band-like excitations of the 3d states. Through theory, these results are used to explore the Ligand hole component in the electronic structure and to conclude that the ground state has a large fraction of oxygen holes.

This work represents a nice piece of science and the authors have addressed my concerns. The paper is now ready for publication in Nature Communications.

[Our reply]: We thank Reviewer #1 once more for his/her reading of our manuscript and for his/her recommendation for publication.

Reviewer #3 (Remarks to the Author):

I am quite satisfied with the authors response to the referees. The experiments reveal important features of the electronic structure and physics of nickelates, bringing a substantial advance to the field of correlated oxides.

[Our reply]: We thank Reviewer #3 for his/her second reading of the manuscript and for acknowledging the thorough character of our response and the overall importance of our work.